# Breast Cancer in Men and Quality of Life: A Systematic Review

**DOI:** 10.3390/cancers17193096

**Published:** 2025-09-23

**Authors:** Milena Giovanna Guarinoni, Paolo Carlo Motta

**Affiliations:** Department of Medical and Surgical Specialties, Radiological Sciences, and Public Health (DSMC), University of Study of Brescia, 25121 Brescia, Italy; paolo.motta@unibs.it

**Keywords:** breast cancer, cancer survivors, health-related quality of life, quality of life, men, nursing, review

## Abstract

Understanding quality of life is important to improve symptom relief, care, and patient rehabilitation. Data suggest the incidence of metastatic breast cancer in men appears to have been increasing over the past 10 years. Although male breast cancer cases are rare, nurses should be able to discern the best possible management of these patients to improve their quality of care. This study aimed to analyze, through a systematic literature review, whether there are studies assessing quality of life in men with male breast cancer. Only 6 articles met the inclusion criteria. Quality of life assessment tools were heterogeneous, and it was not possible to reach a generalizable result.

## 1. Introduction

Quality Of Life (QOL) is a complex concept that is interpreted and defined in a number of ways within and between various disciplines [1]. Although the concept of QOL is extensively used, it has been difficult to define it [2].

The World Health Organization (WHO) defines the QOL as “An individual’s perception of their position in the in the life in the context of the culture in which they live and in relation to their goals, expectations, standards and concerns” [3]. Moreover, the term health-related quality of life (HRQOL) is often described as: “A term referring to the health aspects of quality of life, generally considered to reflect the impact of disease and treatment on disability and daily functioning; it has also been considered to reflect the impact of perceived health on an individual’s ability to live a fulfilling life. However, more specifically HRQOL is a measure of the value assigned to duration of life as modified by impairments, functional states, perceptions and opportunities, as influenced by disease, injury, treatment and policy” [4].

Understanding quality of life (QOL) is essential for enhancing symptom management, patient care, and rehabilitation [1]. Patient-reported QOL outcomes can highlight issues which prompt adjustments in treatment and care or reveal therapies that offer limited benefit. QOL assessments also help to identify a broad range of challenges that patients may face, providing valuable information that can be shared with future patients to help them anticipate and understand the potential impact of their illness and its treatment. Furthermore, patients who are cured or long-term survivors may continue to experience problems well after treatment ends—issues that might go unnoticed without QOL evaluation. QOL also plays a critical role in clinical decision-making, serving as a predictor of treatment success and overall prognosis. Notably, QOL has been demonstrated to be a strong predictor of survival [5].

Reductions in QOL are linked to how patients respond to their disease and its treatment, and these reductions can adversely affect survival [5,6]. Consequently, QOL stands as one of the most crucial patient-reported outcomes in both clinical practice and research [7].

The perception of QOL appears to be different depending on gender and, in fact, it would seem that different instruments should be used depending on gender since men would perceive the same item differently from women [8].

A study which aimed to evaluate the characteristics of QOL papers in medicine and health research, based on a sample of 163 articles, has found that the most prevalent patient groups studied were those with cancer [1].

Gender differences in QOL have been reported in several population-based studies [9,10,11,12], as well as in research spanning various chronic conditions, such as cancer [13,14,15,16,17,18,19,20,21,22,23,24]. Consistently, women generally report a lower QOL than men in at least one evaluated domain [7]. These observed differences suggest that the characteristics predicting QOL are likely distinct for women and men. A deeper understanding of these predictive factors would enable clinicians to identify patients at greater risk of diminished QOL and to implement gender-specific interventions aimed at maintaining or enhancing patient QOL.

Breast cancer (BC) is commonly perceived as a disease that predominantly affects women, due to its high prevalence among the female population [25]. However, male breast cancer (MBC) accounts for approximately 0.5% to 1% of all breast cancer cases worldwide [26]. Despite its rarity, the incidence of BC in men appears to be increasing, with data indicating a rise from 7.2% to 10.3% over the past decade [27]. The low incidence often results in delayed diagnosis, as men are less likely to seek timely medical attention or access appropriate professional and social support. This, in turn, may contribute to greater psychological distress, including body image concerns [28].

While breast cancer has similar biological origins in men and women from an oncological standpoint, the disease in men has been socially disassociated from that in women. This distinction overlooks fundamental differences in how the disease progresses in each gender. 

A common issue is that men are often diagnosed later because their breast symptoms are frequently misdiagnosed as gynecomastia. This delay allows tumors to advance, presenting as larger masses with more extensive lymph node involvement [29]. Therefore, men are more likely to be diagnosed at an advanced stage, leading to worse prognoses and lower survival rates [29,30]. Specifically, the overall survival rate for men with breast cancer stands at 82.8%, compared to 88.5% for women [31].

Despite the rarity of male breast cancer (MBC), nurses must be equipped to manage these patients effectively to enhance their quality of care [32]. A lack of awareness surrounding male breast cancer (MBC), the specific needs of men diagnosed with the disease, and the significant impact MBC has on their lives and quality of life (QOL) places this population at a considerable disadvantage [33]. This gap in understanding may negatively influence the quality of nursing care provided. Therefore, gaining insight into the QOL perceptions of men with breast cancer is crucial for nurses to develop and deliver more comprehensive and holistic care tailored to this patient group [34].

The difference in breast cancer experience and the gender difference in the perception of QOL described above lead to the statement that the quality of life in people affected by BC should also be studied in relation to gender. Although it is known that in the literature there are many studies that examine the QOL in women affected by BC [34,35,36,37,38,39], little is known about the state of study of the QOL in males affected by BC.

The present study aims to analyze, through a systematic review of the literature, whether there are studies that aim to evaluate the quality of life in men affected by BC.

## 2. Materials and Methods

A systematic literature review was conducted between January and March 2025, with the aim of investigating whether the quality of life (QOL) of men diagnosed with breast cancer has been the subject of empirical study.

### 2.1. Research Strategy

We followed the Preferred Reporting Items for Systematic Reviews and Meta-Analyses (PRISMA) guidelines [40]. This study was registered in PROSPERO, registration number is CRD420251016005.

The Web of Science, PubMed, Cinhal, Embase, Cochrane, and Wiley databases were explored. A combination of keywords was utilized in the queries for each database; for PubMed, this was:

(“Breast Neoplasms, Male”[Mesh] OR “Breast Neoplasm, Male” OR “Male Breast Neoplasm” OR “Neplasm, Male Breast” OR “Breast Tumors, Male” OR “Breast Tumor, Male” OR “Male Breast Tumor” OR “Male Breast Tumors” OR “Tumor, Male Breast” OR “Tumors, Male Breast” OR “Male Breast Neoplasms” OR “Neoplasms, Breast, Male” OR “Neoplasms, Male Breast” OR “Tumors, Breast, Male” OR “Male Breast Cancer” OR “Cancer, Male Breast” OR “Breast Cancer, Male” OR “Breast Carcinoma, Male” OR “Carcinoma, Male Breast” OR “Male Breast Carcinoma”) AND (“Quality of Life”[Mesh] OR “Quality of Life” OR “Life Quality” OR “Health-Related Quality Of Life” OR “Health Related Quality Of Life” OR “HRQOL”) 1731 il 20 March 2025.

The keywords were searched for, primarily in the title and abstract, or in the full text if title and abstract searching was not an option, depending on the capabilities of each database.

### 2.2. Inclusion Criteria for Studies

We included primary literature studies utilizing quantitative, qualitative, and/or mixed designs that investigated the Quality of Life (QOL) of men diagnosed with malignant breast cancer. This broad scope covered all histological and morphological types, as well as all TNM stages, and had no time or language constraints. Excluded sources were secondary studies, dissertations, theses, lectures, recommendations, and books.

### 2.3. Data Extraction (Selection and Coding)

Independent screening of titles and abstracts was performed by the two authors. Subsequently, they independently reviewed the full texts to verify adherence to the inclusion criteria. As no disagreements emerged between the two authors, the involvement of a third party was unnecessary. The full text for all relevant studies was directly acquired from the surveyed libraries, thereby precluding the need to contact the respective authors. Extracted data were categorized according to the following variables: year of publication, author(s), article title, country of study, study purpose, and conclusions (see Table 1).

## 3. Results

Of the 2410 references retrieved from the queried databases, duplicate removal reduced the number to 731 articles for screening. After excluding records based on title and abstract, 46 full-text articles were assessed, ultimately resulting in the inclusion of 6 studies that met the eligibility criteria (Figure 1).

The characteristics of the six studies included in the review are summarized in Table 1.

The studies included in this systematic review were conducted between the years 2012 and 2023.

Andrykowski’s 2012 study [41] utilized data from the 2009 Behavioral Risk Factor Surveillance System (BRFSS), an annual national telephone survey monitoring health conditions and risk behaviors across the United States and its territories. The study identified 66 male breast cancer (MBC) cases, who were then matched to 198 BRFSS men with no history of cancer (control group) based on age, education, and minority status.

Comparisons between the MBC and control groups regarding physical and mental health status and health behaviors were conducted using *t*-tests and logistic regression analyses. The MBC group reported significantly poorer physical and mental health compared to the controls. Specifically, male breast cancer survivors were significantly more likely (*p* < 0.05) to be obese (Odds Ratio [OR] = 2.41) and reported more physical comorbidities (Effect Size [ES] = 0.45), greater physical activity limitations (OR = 3.17), lower life satisfaction (ES = 0.41), and poorer general health (ES = 0.40). Furthermore, they reported more days in the past month with poor mental health (ES = 0.49) and poor physical health (ES = 0.29). In contrast, the MBC and control groups demonstrated similarity in current health behaviors, including tobacco and alcohol use, diet, exercise, and healthcare utilization.

In their 2012 study, Kowalski et al. [42] sought to evaluate health-related quality of life (HRQOL) among male breast cancer patients. The authors analyzed data from a cohort of 20,673 individuals diagnosed with primary breast cancer, including 84 men, who completed a post-discharge questionnaire. HRQOL was assessed using the Short Form-36 (SF-36) instrument, alongside demographic variables such as age, sex, education, native language, insurance status, and relationship status. Clinical characteristics, including cancer stage, treatment modality (partial versus radical mastectomy), and tumor location, were also recorded. Statistical analyses involved *t*-tests and regression models to compare HRQOL outcomes between male and female patients. The results indicated that male breast cancer patients reported significantly higher HRQOL scores on seven of the eight SF-36 subscales relative to female patients, suggesting comparatively better perceived health status in this population. These subscales included physical functioning, physical and emotional role functioning, bodily pain, vitality, social functioning, and mental health. However, when compared to male reference populations (general male population, men aged 61–70 years, and a general cancer population), male breast cancer patients obtained lower HRQOL scores, with significant differences particularly in emotional and physical role functioning. Based on these findings, the study concluded that male breast cancer patients may benefit from early interventions specifically targeting role functioning, as this aspect appears severely impaired compared to the general male reference population.

A brief report by Ruddy et al., published in 2013 [26], investigated the experiences of men with stage 0–4 breast cancer by recruiting 42 participants through an online survey. This comprehensive survey included established instruments such as the Expanded Prostate Cancer Index Composite (EPIC) hormonal/sexual scales, the Hospitalized Anxiety and Depression Scale (HADS), and the Functional Assessment of Breast Cancer Therapy (FACT-B), in addition to questionnaires covering sociodemographic, disease-related, genetic, and fertility aspects. The findings highlighted several critical areas of concern for male breast cancer survivors. Notably, forty percent of respondents reported “very poor” ability to have sexual intercourse in the preceding four weeks. For men undergoing endocrine therapy, the mean EPIC sexuality score was 43.5 (SD 20.1) and the mean hormonal function score was 80.3 (SD 15.6), with similar scores observed in men not receiving endocrine therapy (46.0, SD 34.1, and 82.9, SD 15.8, respectively). While only a minority of participants met standard criteria for anxiety and depression, the mean FACT-B score of 111.1 (SD 19.9), completed by 90% of participants, indicated an overall impact on their quality of life. The study also delved into genetic and fertility issues: thirty-one participants (74%) recalled being referred for genetic counseling, and 25 subsequently underwent testing, with three (12% of those tested) found to carry a cancer-causing mutation. Regarding fertility, ten participants (24%) stated they had no biological children, yet only two (5%) expressed a desire to have children in the future, and just one (2%) had banked sperm prior to treatment (It was noted that one participant did not answer genetic testing questions, though all responded to fertility inquiries). Ultimately, the report concluded that despite the prevalent model of care for men with breast cancer being based on that for women, the disease experience and survivorship concerns are demonstrably different for men.

In 2022, Majdouline El Fouhi [43] conducted the first study in Morocco to evaluate the quality of life (QOL) in male breast cancer patients. This retrospective study aimed to assess the health-related quality of life (HRQOL) of male breast cancer patients treated at the University Hospital of Casablanca over a six-year period (2012–2018). A total of 21 male patients were included, with relevant demographic, clinical, and pathological data extracted from oncology center medical records. All participants had undergone curative cancer therapy. The overall QOL was reported as moderate, with a mean score of 50 ± 21.73. Domain-specific results indicated mean scores of 54.60 ± 27.85 for physical functioning, 56.34 ± 31.94 for emotional functioning, and 75.39 ± 17.96 for social functioning. Symptom burden included fatigue (mean 61.9 ± 31.34), nausea/vomiting (43.88 ± 36.15), pain (40.47 ± 29.61), dyspnea (36.50 ± 31.45), insomnia (41.26 ± 37.86), loss of appetite (46.03 ± 34.11), constipation (experienced by 20.8% of patients, mean 44.44 ± 79.11), and diarrhea (28.57 ± 42.53). Financial difficulties were reported with a mean score of 61.90 ± 32.12. The study concluded that QOL was generally better than expected and improved following treatment. Notably, emotional and social functioning were well preserved despite a lack of significant improvement in physical functioning, likely attributable to the presence of an effective social support system. The study also emphasized the need for public education campaigns targeting men at higher risk of breast cancer to reduce diagnostic delays. Additionally, it highlighted that future quality of life research in male breast cancer patients should give greater attention to disease-related issues, treatment side effects, and sexual functioning.

The study described in the article by Avila et al. (2023) [44] also used an online survey of male breast cancer patients from June to July 2022. Participants were surveyed regarding disease characteristics, treatments received, and side effects related to both the disease and its treatment. A total of 127 responses were analyzed, with a mean participant age of 64 years (range 56–71 years). Among the respondents, 91 (71.7%) reported experiencing late effects secondary to the tumor or anticancer treatment. Of these, 71 men (78%) reported physical symptoms, with fatigue being the most commonly cited concern (33%). Psychological effects were reported by 51 participants (56%), with fear of recurrence identified as the predominant concern (28.1%). Treatment-related symptoms included hot flashes in 63 participants (49.6%), while 69 men (54.3%) reported feeling less masculine due to their illness or treatment. A majority of respondents (78.7%, n = 100) indicated that their treatment adversely affected their sexual interest. Additionally, 75 participants (61%) experienced varying degrees of distressing hair loss, and 70 (55.6%) reported persistent pain at the surgical scar site beyond the typical postoperative recovery period. Swelling in the arm or hand was reported by 42 participants (33.1%), and 66 individuals (52.8%) experienced some degree of impaired arm or shoulder mobility attributed to surgery. Participants primarily reported fatigue as their most concerning physical symptom and fear of recurrence as their most concerning psychological symptom. Financial hardship affected nearly half of the respondents (n = 63, 49.6%), manifesting as reduced work hours, job or health insurance loss, inadequate disability insurance, and other forms of financial strain. Encouragingly, 108 men (85%) indicated they did not have to change their recommended treatment plans due to these financial challenges. The research highlighted that men frequently contend with various long-term side effects from breast cancer treatments. Clinicians should proactively address concerns such as lymphedema, limited arm and shoulder mobility, sexual dysfunction, and hair loss with male patients, given their potential to cause distress and diminish quality of life.

The International Male Breast Cancer Program established a prospective registry (EORTC10085) for men with breast cancer (BC) across all stages [45]. This comprehensive initiative included a companion quality of life (QOL) study (Schröder et al., 2023) [45]. At the time of BC diagnosis, participants completed the EORTC QLQ-C30 and BR23 questionnaires, both specifically adapted for men. It is important to note that higher scores on functioning and global health/QOL scales reflect better functioning and higher QOL, whereas elevated scores on symptom-focused measures indicate a greater burden of symptoms or problems. For comparative analyses, baseline EORTC data from healthy men and women with BC were incorporated. Out of 422 men who consented to participate, 363 were ultimately evaluable. The median age of these participants was 67 years, and the median interval between diagnosis and survey completion was 1.1 months. Within the evaluable cohort, 114 men (45%) presented with early node-positive disease, while 28 (8%) were diagnosed with advanced disease. The mean baseline global health score was 73 (SD: 21), which was notably higher than the corresponding baseline score reported for female breast cancer patients (62, SD: 25). The most common symptoms among male breast cancer patients included fatigue (mean score 22, SD: 24), insomnia (21, SD: 28), and pain (16, SD: 23). In contrast, female patients reported higher symptom burden, with mean scores of 33 (SD: 26), 30 (SD: 32), and 29 (SD: 29) for these symptoms, respectively. The mean sexual activity score among men was 31 (SD: 26), with reduced sexual activity observed in older patients and those with advanced disease. This significant study suggests that quality of life and symptom burden in male breast cancer patients are not worse—and may in fact be better—than those reported by female patients. The study further concluded that future analyses of the impact of treatment on symptoms and quality of life over time may support the personalization of management of male BC.

## 4. Discussion

The primary objective driving this systematic review was to meticulously ascertain the presence and nature of the existing scientific literature concerning the Quality of Life (QOL) among men diagnosed with breast cancer. This particular area of inquiry holds considerable and often underestimated significance, especially given the evolving epidemiology of male breast cancer (MBC). Recent data from 2019 in the United States, for instance, reported 2670 new MBC cases and 500 associated deaths [46]. Furthermore, while conventionally seen as a rare disease, MBC accounts for approximately 1% of all breast cancer diagnoses globally, a percentage that has demonstrated a discernible upward trend in recent years [47]. Despite these escalating figures, and despite a 5-year survival rate for MBC (77.6%) that remains notably lower than that for female breast cancer (86.4%) [46], the pervasive societal perception continues to erroneously frame breast cancer as predominantly a “women’s problem.” This deep-seated gender bias in public consciousness, unfortunately, extends into the research domain, leading to a disproportionate allocation of investigative efforts almost exclusively towards female patients. Consequently, while the literature abounds with studies focusing on the female gender, the unique challenges and concerns pertinent to the same pathology in male patients are, far too often, overlooked or marginalized. Therefore, the central hypothesis underpinning this review was a clear expectation to uncover a significant shortage of dedicated scientific works addressing QOL in men with breast cancer.

Our systematic search, conducted across publications spanning from 2012 to the present, regrettably confirmed this initial hypothesis. We identified merely six primary studies that had specifically ventured into the realm of QOL for men affected by breast cancer. Furthermore, among this extremely limited collection, only one study, a very recent publication from 2023, incorporated a substantial and geographically diverse cohort of participants. This solitary study thus offered the most compelling, albeit still nascent, opportunity for reflecting on relatively generalizable data concerning male QOL in this context. The profound importance of maintaining a good QOL in cancer care is unequivocally established within oncology. It is recognized as an integral component in the holistic process of symptom management, patient care, and rehabilitation [1]. Beyond its immediate impact on well-being, QOL plays a pivotal role in informing medical decision-making and has even been consistently identified as a significant predictor of survival [5]. As a direct consequence of its multifaceted importance, QOL stands as one of the most crucial patient-reported outcomes (PROs), diligently monitored and assessed in both routine clinical practice and rigorous scientific research [7].

Perhaps owing to this well-demonstrated importance, the endeavor to measure QOL, particularly disease-related QOL, has spurred the creation and validation of numerous assessment tools [48]. Nevertheless, a notable challenge, as highlighted by the present review, lies in the inherent heterogeneity of variables encompassed within these validated instruments and, subsequently, across the studies that utilize them. This pervasive variability frequently complicates, and often renders impossible, the meaningful synthesis of results across disparate works or the conduct of robust meta-analyses that could otherwise provide unequivocal answers. Perhaps precisely because of this widely demonstrated importance, the effort to measure quality of life, particularly disease-related quality of life, has spurred the creation and validation of numerous assessment tools [48]. However, a notable challenge, as highlighted by this review, lies in the intrinsic heterogeneity of the variables included in these validated tools and, consequently, in the studies that use them. This pervasive variability often complicates, and often makes impossible, the meaningful synthesis of findings from different studies or the conduct of robust meta-analyses that might otherwise provide unequivocal answers. The development of a single, validated instrument would be crucial. The EORTIC, for example, provides a core questionnaire, the QLQ-C30, which assesses general cancer-related quality of life, and then integrates it with specific modules for different cancer types. This method allows both comparability across different cancer populations and the ability to capture unique and disease-specific issues. For male patients with breast cancer (MBC), this modular approach is crucial. The EORTC is developing a specific module for metastatic breast cancer (MBC), which would include elements related to body image and masculinity, psychosocial support, and treatment-related side effects. This development directly addresses the limitations of current tools, ensuring that future clinical trials and patient care can accurately measure and address the specific quality of life issues in metastatic breast cancer (MBC).

An additional factor contributing to the incomparability of the identified studies is their divergent methodological approaches to comparative analysis. Some investigations chose to compare the QOL of men with breast cancer directly against that of women afflicted by the same pathology [26,41]. In contrast, other studies developed their comparisons with a healthy male population [42], while a third subset of studies refrained from making any direct comparisons with other groups altogether [43,44,45].

Despite these methodological variances, certain consistent insights emerged. Studies that compared the QOL of men and women with breast cancer, for instance, consistently indicated that men developed greater deficits in both physical and mental health when facing the same pathology compared to their female counterparts [41]. Furthermore, a significant symptomatic burden was observed within the sexual and hormonal domains, consistent with a broader compromise in overall QOL. These findings compellingly underscore that, even though the approach to male breast cancer care has historically been modeled on that for women, men’s specific experiences with the disease and their concerns during survivorship are demonstrably unique. A 2024 retrospective review found that 22% of male patients taking tamoxifen experienced sexual dysfunction or loss of libido, and 23% of patients who discontinued treatment did so specifically because of this side effect [49]. This underscores that sexual dysfunction is not just a passing mention but a concrete reason men stop a vital treatment, which is an implicit criticism of the lack of attention this issue receives. This distinctiveness necessitates a type of care that is more precisely attuned to their particular needs [26].

Even in instances where the comparison was drawn against a healthy male population, the variable pertaining to role functioning was repeatedly shown to be severely compromised. This particular finding strongly suggests the imperative for early and targeted interventions specifically designed to prevent or mitigate such impairment in male breast cancer patients [42].

In the third category of studies, where no comparisons with other groups were undertaken, a general consensus emerged regarding the assessment of QOL in this male patient population: it was largely deemed to be good. However, within this overall positive assessment, certain specific disorders, such as sexual dysfunction, loss of appetite, fatigue, and insomnia, were still described. These, nonetheless, were often considered better than initially expected [44], with an overarching trend indicating that QOL generally improved after treatment [45].

Crucially, regarding the aspect of prevention, the literature consistently emphasizes that public education campaigns should be meticulously directed towards men identified as being at the highest risk. The objective here is to significantly reduce the critical time interval that frequently elapses between the initial onset of symptoms and the subsequent medical consultation [43]. This diagnostic delay is, unfortunately, a very common problem routinely detected in numerous epidemiological studies specifically related to MBC [29,30]. Based on what has been described in clinical practice, specific interventions should be implemented, such as financial counseling, sexual health programs, and lymphedema management.

In conclusion, the current state of the literature, characterized by the extremely small sample sizes of the identified studies and the pervasive heterogeneity of the tools and variables employed for data collection, undeniably precludes the provision of a definitive and comprehensive answer to the complex question regarding disease-related QOL in men affected by breast cancer. Other studies from 2025, such as a review on the current landscape of male breast cancer, also highlight the “limited data guiding treatment strategies” and the “urgent need for male-specific clinical trials” [50]. This reinforces the idea that the article you provided is an accurate representation of a major gap in science. To effectively bridge this substantial knowledge gap and advance our understanding, future research endeavors should ideally commence with in-depth qualitative studies. Such studies would aim to thoroughly analyze and comprehend the most critical variables and unique needs from the direct perspective of these patients. Subsequently, these invaluable qualitative insights could then be rigorously verified through larger-scale quantitative studies. This methodical, two-pronged approach would ultimately facilitate the development of personalized and truly tailored care strategies that cater to the specific needs of men with breast cancer, needs that, fundamentally, cannot be superficially equated to those of women. We also should prioritize longitudinal studies to track how QOL evolves throughout a man’s breast cancer treatment and into survivorship, providing a more complete picture than a single point in time. Additionally, integrating qualitative work is essential to explore the unmet needs and hidden struggles of MBC patients, such as the profound impact on body image and crises of masculinity.

Limits: The lack of rigorous quality assessment in QOL studies limits our confidence in their findings, making it difficult to determine with certainty how reliable and generalizable the results are.

Furthermore, many studies suffer from “survivorship bias”, as they only include people who have survived, ignoring the quality of life experiences during the most difficult treatment phases and for those who did not survive, thus offering an incomplete picture.

## 5. Conclusions

A critical examination of the literature reveals a significant scarcity of studies dedicated to the quality of life (QOL) in male breast cancer (MBC) patients. This is primarily because breast cancer is rare in men, accounting for less than 1% of all cases. As a result, research and treatment strategies have historically been extrapolated from studies on female breast cancer (FBC), often overlooking the unique physical and psychosocial challenges faced by men. This gap in knowledge underscores a pressing need for more focused research.

The majority of existing research on MBC QOL is limited, often consisting of small-scale retrospective studies or single-institution analyses. While these provide valuable insights, they do not capture the evolving nature of QOL over the disease trajectory. Therefore, there is a clear and urgent need for longitudinal studies that can track changes in physical, emotional, and social well-being from diagnosis through treatment and into survivorship. Such studies would allow for a deeper understanding of the long-term impacts of MBC and its treatments, helping to identify critical time points for intervention and providing a more comprehensive picture of the patient experience. This approach is essential for developing evidence-based, male-specific interventions and support systems that can truly enhance the QOL for men living with and beyond breast cancer.

## Figures and Tables

**Figure 1 cancers-17-03096-f001:**
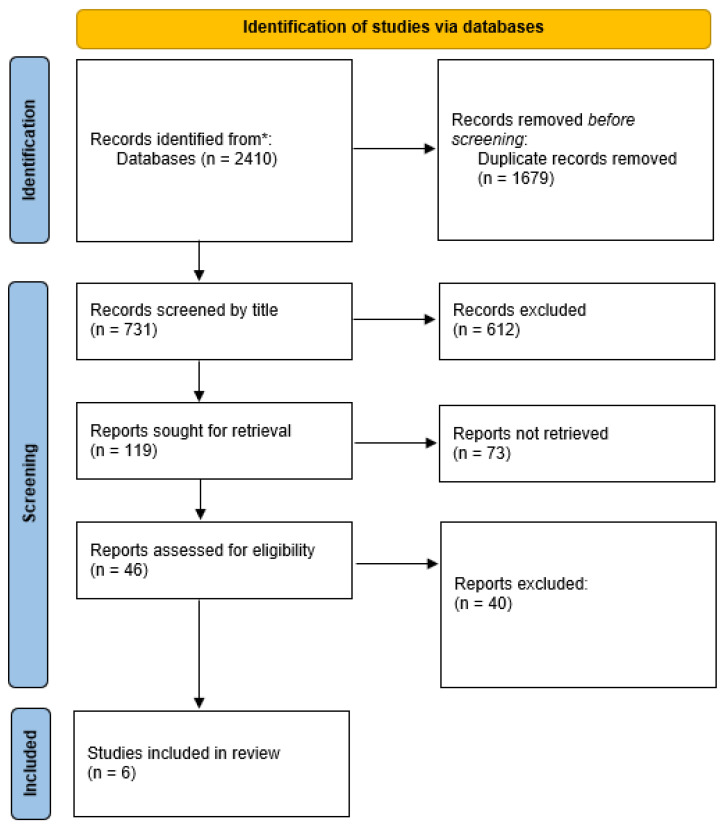
PRISMA flow diagram.

**Table 1 cancers-17-03096-t001:** Characteristics of the six studies included in the review.

Year	Author	Journal	Title	Country	Purpose	Study Type	Population	Conclusions
2012	Andrykowski MA [41]	Psychooncology	Physical and mental health status and health behaviors in male breast cancer survivors: a national, population-based, case-control study	USA	To examine current physical and mental health status and health behaviors in male breast cancer survivors	CASE-CONTROL	66 MBC 198 male with no history of cancer	The diagnosis and treatment of male breast cancer may be associated with clinically important and long-term deficits in physical and mental health status, deficits which may exceed those evidenced by long-term female breast cancer survivors
2012	Kowalski C et al. [42]	Breast Cancer Res Treat	Health-related quality of life in male breast cancer patients	Germany	The aim of the study was to compare the quality of healthcare provided in the individual breast centers as perceived by the patients. A secondary objective was to describe the HRQOL of the patients.	Descriptive	84 MBC WBC and male without cancer	Male breast cancer patients may need early interventions that specifically target role functioning, which is severely impaired compared to the male reference population.
2013	Ruddy KJ et al. [26]	Breast	Quality of life and symptoms in male breast cancer survivors.	USA	To learn more about symptoms, quality of life, distress, and fertility and genetic issues in male breast cancer survivors.	Survey	42 MBC	With mean EPIC Sexual and Hormonal scores of 44.5 and 81.3, patients reported a considerable symptom burden. The mean FACT-B score of 111.1 also pointed to an impaired overall quality of life. While care for men with breast cancer often mirrors that for women, men’s unique experiences with the disease and their specific concerns during survivorship remain.
2022	Fouhi ME et al. [43]	Pan Afr Med J	Quality of life and epidemiological profile of male breast cancer treated at the university hospital of Casablanca, Morocco	Morocco	To investigate health-related quality of life (HRQOL) and epidemiological profile of Moroccan male breast cancer patients at the university hospital of Casablanca Ibn Rochd	Descriptive	21 MBC	Regarding QOL in this population, it appears to be better than expected and QOL generally improves after treatment. As for prevention, public education should be oriented toward men at higher risk in order to reduce the time between onset of symptoms and consultation.
2023	Avila J et al. [44]	Breast Cancer Res Treat	Treatments for breast cancer in men: late effects and impact on quality of life.	USA	To survey this particular group of patients about their experience with BC.	Survey	127 MBC	The study provides critical information on several side effects and late effects that are experienced by male patients with breast cancer. Further research is necessary to mitigate the impact of these effects and improve quality of life in men.
2023	Schröder CP et al. [45]	Oncologist	Quality of Life in Male Breast Cancer: Prospective Study of the International Male Breast Cancer Program	International	To assess quality of life (QOL) and symptom burden, which have been woefully understudied in male BC.	Survey	363 MBC WBC	This large prospective registry substudy demonstrates that overall QOL is good in men who were recently diagnosed with breast cancer, but some still suffer appetite loss, fatigue, and insomnia. Sexual functioning may also be an issue.

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
