# Peer review of "Breast Cancer in Men and Quality of Life: A Systematic Review"

_cancers, 2025, doi:10.3390/cancers17193096_

Round 1
Reviewer 1 Report
Comments and Suggestions for Authors
"Breast Cancer in Men and Quality of Life: A Systematic Review" is an important starting point, but it has several limitations inherent to the lack of research on the topic. My critique is based on the need to contextualize its findings and delve into aspects that, while mentioned, cannot be fully developed due to the scarcity of data.
-
The lack of studies as its main limitation: The systematic review found only 6 primary studies, an extremely low number. This is less a criticism of the article itself and more a reflection of the current state of research. Therefore, the article serves more as a wake-up call than a document with firm conclusions. Other studies from 2025, such as a review on the current landscape of male breast cancer, also highlight the "limited data guiding treatment strategies" and the "urgent need for male-specific clinical trials" (PMC12239999). This reinforces the idea that the article you provided is an accurate representation of a major gap in science.
-
Heterogeneity in quality of life assessment: The article mentions that the studies used a variety of assessment tools, which prevents a generalizable conclusion. This is a valid and necessary criticism. A 2025 review on ResearchGate on the same topic, which included 3 studies, also points out this limitation and concludes that the "limited number of studies... highlights the need for further research" (ResearchGate, 2025). The lack of a standard for measuring quality of life makes the results difficult to compare and apply in clinical practice.
-
Lack of depth on specific quality of life issues: While your article mentions sexual dysfunction, fatigue, and insomnia as common problems, it cannot detail the extent of these problems or their causes due to the lack of data.
-
Sexual Dysfunction: A 2024 retrospective review found that 22% of male patients taking tamoxifen experienced sexual dysfunction or loss of libido, and 23% of patients who discontinued treatment did so specifically because of this side effect (PMC3493137). This underscores that sexual dysfunction is not just a passing mention but a concrete reason men stop a vital treatment, which is an implicit criticism of the lack of attention this issue receives.
-
Psychosocial Impact: Your article notes that some studies compared the quality of life of men with breast cancer to that of women and healthy men. However, searches in English reveal a deeper critique: men often experience "emasculation," isolation, and shame, as breast cancer is perceived as a "woman's disease." A 2019 qualitative study even quotes a patient asking, "My biggest problem was how to tell my wife that I have a woman's disease? Because I thought maybe you're not a real man, perhaps half woman?" (PMC7098286). The article you provided does not delve into this psychosocial dimension, which is a crucial part of quality of life.
-
Author Response
Dear Sir,
Thank you for taking the time to review my work. Regarding your valuable comments, I would like to inform you that
-
The lack of studies as its main limitation: The systematic review found only 6 primary studies, an extremely low number. This is less a criticism of the article itself and more a reflection of the current state of research. Therefore, the article serves more as a wake-up call than a document with firm conclusions. Other studies from 2025, such as a review on the current landscape of male breast cancer, also highlight the "limited data guiding treatment strategies" and the "urgent need for male-specific clinical trials" (PMC12239999). This reinforces the idea that the article you provided is an accurate representation of a major gap in science.
RESPONSE: the citation of the review has been added
Heterogeneity in quality of life assessment: The article mentions that the studies used a variety of assessment tools, which prevents a generalizable conclusion. This is a valid and necessary criticism. A 2025 review on ResearchGate on the same topic, which included 3 studies, also points out this limitation and concludes that the "limited number of studies... highlights the need for further research" (ResearchGate, 2025). The lack of a standard for measuring quality of life makes the results difficult to compare and apply in clinical practice.
RESPONSE: I think this is just a consideration that I fully share.
-
Lack of depth on specific quality of life issues: While your article mentions sexual dysfunction, fatigue, and insomnia as common problems, it cannot detail the extent of these problems or their causes due to the lack of data.
-
Sexual Dysfunction: A 2024 retrospective review found that 22% of male patients taking tamoxifen experienced sexual dysfunction or loss of libido, and 23% of patients who discontinued treatment did so specifically because of this side effect (PMC3493137). This underscores that sexual dysfunction is not just a passing mention but a concrete reason men stop a vital treatment, which is an implicit criticism of the lack of attention this issue receives.
-
Psychosocial Impact: Your article notes that some studies compared the quality of life of men with breast cancer to that of women and healthy men. However, searches in English reveal a deeper critique: men often experience "emasculation," isolation, and shame, as breast cancer is perceived as a "woman's disease." A 2019 qualitative study even quotes a patient asking, "My biggest problem was how to tell my wife that I have a woman's disease? Because I thought maybe you're not a real man, perhaps half woman?" (PMC7098286). The article you provided does not delve into this psychosocial dimension, which is a crucial part of quality of life.
-
RESPONSE: the citation of the article PMC3493137 has been added
For the psychosocial impact, I believe it truly deserves further study in a new study.
Reviewer 2 Report
Comments and Suggestions for Authors
The authors reported their work named "Breast Cancer in Men and Quality of Life: A Systematic Review". Here are my comments.
Strengths
- Relevance and Novelty:
- Addresses a critical gap in oncology literature by focusing on QOL in male breast cancer (MBC), a rare and understudied population.
- Highlights rising MBC incidence (7.2%→10.3% over 10 years) and poorer survival (82.8% vs. 88.5% in women), justifying urgent attention.
- Methodological Rigor:
- Follows PRISMA guidelines, searches six major databases (PubMed, Cochrane, etc.), and uses clear inclusion/exclusion criteria.
- Independent screening by two authors minimizes bias.
- Key Findings:
- Confirms severe QOL impacts: role functioning deficits, sexual dysfunction, psychological distress (fear of recurrence), and financial toxicity.
- Identifies gender-specific disparities: Men report worse physical/mental health than controls but better global QOL than female patients.
- Notes QOL improvement post-treatment but highlights persistent issues (fatigue, insomnia).
- Clinical Implications:
- Advocates for gender-tailored interventions (e.g., early support for role functioning, sexual health).
- Urges public education to reduce diagnostic delays linked to misdiagnosis (e.g., gynecomastia).
Weaknesses and Limitations
- Limited Evidence Base: Only 6 studies included (2012–2023), with small samples (e.g., n = 21 in Fouhi et al.). Geographic bias: Most studies from the US/Europe; only one (Morocco) from Africa.
- Methodological Heterogeneity: Diverse QOL tools (SF-36, FACT-B, EORTC QLQ-C30) preclude meta-analysis or unified conclusions. Varied comparators: Some studies used female BC cohorts, others healthy males, or no controls.
- Quality Assessment Gap: No critical appraisal (e.g., risk of bias) of included studies, weakening conclusions.
- Overlooked Intersectional Factors: Socioeconomic/cultural determinants (e.g., stigma, masculinity norms) are discussed but not systematically analyzed.
Recommendations for Revision
- Expand Discussion: Delve deeper into why QOL tools are heterogeneous and propose standardization (e.g., EORTC male-specific modules). Address how MBC stigma (mentioned in references) directly impacts QOL.
- Strengthen Limitations Section: Explicitly state that lack of quality assessment limits confidence in findings. Acknowledge survivorship bias (studies include survivors; QOL during active treatment is underrepresented).
- Future Research Directions: Prioritize longitudinal studies to track QOL changes during/after treatment. Include qualitative work to explore unmet needs (e.g., body image, masculinity crises).
- Clinical Practice Implications: Recommend specific interventions: e.g., financial counseling, sexual health programs, and lymphedema management.
Author Response
Thank you very much for taking the time to review this manuscript.
1. Expand Discussion: Delve deeper into why QOL tools are heterogeneous and propose standardization (e.g., EORTC male-specific modules). Address how MBC stigma (mentioned in references) directly impacts QOL.
RESPONSE the sentence was insert:
The development of a single, validated instrument would be crucial. The EORTIC, for example, provides a core questionnaire, the QLQ-C30, which assesses general cancer-related quality of life, and then integrates it with specific modules for different cancer types. This method allows both comparability across different cancer populations and the ability to capture unique and disease-specific issues. For male patients with breast cancer (MBC), this modular approach is crucial. The EORTC is developing a specific module for metastatic breast cancer (MBC), which would include elements related to body image and masculinity, psychosocial support, and treatment-related side effects. This development directly addresses the limitations of current tools, ensuring that future clinical trials and patient care can accurately measure and address the specific quality of life issues in metastatic breast cancer (MBC).
2. Strengthen Limitations Section: Explicitly state that lack of quality assessment limits confidence in findings. Acknowledge survivorship bias (studies include survivors; QOL during active treatment is underrepresented).
RESPONSE the sentence was insert:
Limits: The lack of rigorous quality assessment in QOL studies limits our confidence in their findings, making it difficult to determine with certainty how reliable and generalizable the results are.
Furthermore, many studies suffer from "survivorship bias", as they only include people who have survived, ignoring the quality of life experiences during the most difficult treatment phases and for those who did not survive, thus offering an incomplete picture.
3. Future Research Directions: Prioritize longitudinal studies to track QOL changes during/after treatment. Include qualitative work to explore unmet needs (e.g., body image, masculinity crises).
RESPONSE the sentence was insert:
We also should prioritize longitudinal studies to track how QOL evolves throughout a man's breast cancer treatment and into survivorship, providing a more complete picture than a single point in time. Additionally, integrating qualitative work is essential to explore the unmet needs and hidden struggles of MBC patients, such as the profound impact on body image and crises of masculinity.
4. Clinical Practice Implications: Recommend specific interventions: e.g., financial counseling, sexual health programs, and lymphedema management.
RESPONSE the sentence was insert:
Based on what's been described in clinical practice, specific interventions should be implemented, such as financial counseling, sexual health programs, and lymphedema management.
Reviewer 3 Report
Comments and Suggestions for Authors
The abstract should follow the PRISMA abstract checklist, indicating which line each item is on and, if any are missing, the reason for its absence.
Although the total number of articles selected (6) addresses a very understudied topic, it is insufficient to draw conclusions. Internal and external validity are widely questioned. No distinction is made between quality of life in patients with or without surgery, or whether the surgery is more or less aggressive. Nor is the effect of social status, etc., studied. The main population is the US. All of this can lead to significant bias in the results, which are not analyzed in depth.
More quantitative, qualitative, or mixed studies will be needed before a systematic review can be conducted and conclusions drawn.
Furthermore, authors should comply with the PRISMA Checklist for Systematic Reviews, indicating the line in which the item is developed or the reason for its absence.
The conclusion that there are not enough studies to draw conclusions is interesting because it opens the door to developing them.
Author Response
The limitations and discussion emphasize that, given the paucity of the material, it is not possible to draw any conclusions. I believe this review is important precisely because it highlights the lack of primary studies on the topic.
The PRISMA checklist has been compiled, but it was not possible to precisely indicate the rows because some items are scattered throughout broad discussions.
Round 2
Reviewer 3 Report
Comments and Suggestions for Authors
We appreciate that the authors followed the referees' instructions.
Methodologically, I think it is correct. The PRYSMA system has been faithfully followed. Although the results are not very consistent due to the scarcity of studies on the topic, I recommend that the conclusions emphasize this scarcity of studies on quality of life in male breast cancer and the need for longitudinal studies.
Author Response
Methodologically, I think it is correct. The PRYSMA system has been faithfully followed. Although the results are not very consistent due to the scarcity of studies on the topic, I recommend that the conclusions emphasize this scarcity of studies on quality of life in male breast cancer and the need for longitudinal studies.
RESPONSE
Thank you for your further encouragement to improve this study. As recommended, I have reworded the conclusions, emphasizing the paucity of research on the topic and the need for longitudinal studies:
Conclusions: A critical examination of the literature reveals a significant scarcity of studies dedicated to the quality of life (QoL) in male breast cancer (MBC) patients. This is primarily because breast cancer is rare in men, accounting for less than 1% of all cases. As a result, research and treatment strategies have historically been extrapolated from studies on female breast cancer (FBC), often overlooking the unique physical and psychosocial challenges faced by men. This gap in knowledge underscores a pressing need for more focused research.
The majority of existing research on MBC QoL is limited, often consisting of small-scale retrospective studies or single-institution analyses. While these provide valuable insights, they do not capture the evolving nature of QoL over the disease trajectory. Therefore, there is a clear and urgent need for longitudinal studies that can track changes in physical, emotional, and social well-being from diagnosis through treatment and into survivorship. Such studies would allow for a deeper understanding of the long-term impacts of MBC and its treatments, helping to identify critical time points for intervention and providing a more comprehensive picture of the patient experience. This approach is essential for developing evidence-based, male-specific interventions and support systems that can truly enhance the QoL for men living with and beyond breast cancer.